# Molecular and Cellular Pathogenesis of Ellis-van Creveld Syndrome: Lessons from Targeted and Natural Mutations in Animal Models

**DOI:** 10.3390/jdb8040025

**Published:** 2020-10-09

**Authors:** Ke’ale W. Louie, Yuji Mishina, Honghao Zhang

**Affiliations:** Department of Biologic and Materials Sciences & Prosthodontics, School of Dentistry, University of Michigan, Ann Arbor, MI 48109-1078, USA; keale@umich.edu

**Keywords:** Ellis-van Creveld syndrome, EVC2, craniofacial, LIMBIN, ciliopathy

## Abstract

Ellis-van Creveld syndrome (EVC; MIM ID #225500) is a rare congenital disease with an occurrence of 1 in 60,000. It is characterized by remarkable skeletal dysplasia, such as short limbs, ribs and polydactyly, and orofacial anomalies. With two of three patients first noted as being offspring of consanguineous marriage, this autosomal recessive disease results from mutations in one of two causative genes: *EVC* or *EVC2/LIMBIN*. The recent identification and manipulation of genetic homologs in animals has deepened our understanding beyond human case studies and provided critical insight into disease pathogenesis. This review highlights the utility of animal-based studies of EVC by summarizing: (1) molecular biology of EVC and EVC2/LIMBIN, (2) human disease signs, (3) dysplastic limb development, (4) craniofacial anomalies, (5) tooth anomalies, (6) tracheal cartilage abnormalities, and (7) EVC-like disorders in non-human species.

## 1. Introduction

First described in 1940, Ellis-van Creveld syndrome (EVC; MIM ID #225500) is a rare, recessive congenital disorder that results in a type of disproportionate dwarfism [1]. Affected individuals represent approximately 1 in every 60,000–200,000 live births with higher occurrence reported in offspring of consanguineous (e.g., first cousin) unions or monotypic populations [2,3]. In addition to short stature (Figure 1A), other conspicuous signs of EVC include extra digits (i.e., polydactyly), dysmorphic faces, and dental anomalies [4]. While none of these directly confer shortened longevity, approximately 60% of patients suffer from underlying cardiac conditions, indicating widespread developmental effects.

Although EVC was known to occur more frequently in certain families and/or communities (e.g., the Pennsylvania Amish), several decades passed before the causative gene(s) (i.e., *EVC* and *EVC2/LIMBIN*) were identified. Mutations in either of these head-to-head genes located on human chromosome 4p were found to contribute to EVC and a dominant but phenotypically milder form called Weyers acrodental dysostosis (also known as Curry–Hall syndrome, WAD; MIM ID #193530)) [3,5]. The primary difference between EVC and WAD is the severity of clinical phenotypes and pattern of inheritance. Though mutations in *EVC* and *EVC2* account for the majority of the patients, there are still patients that present with EVC symptoms despite no mutations in either *EVC* or *EVC2* [6,7]. EVC is an autosomal recessive disorder and is associated with a more severe phenotype, specifically heart abnormalities, which are the main cause of morbidity. Since *EVC* and *EVC2* share limited homology, *EVC* and *EVC2* were named based on a historic timeline of mapping and cloning of causative genes for EVC, and thus they do not form a gene family. In this manuscript, we use both *EVC2* and *LIMBIN* to describe this gene, because pathophysiologic function of this gene was firstly identified as a causative gene of Bovine chondrodysplastic dwarfism and named as *LIMBIN* [8].

Animal models of EVC have been critical for providing insight into disease pathogenesis (detailed in subsequent sections) [9,10,11]. The initial observation was from *EVC2/LIMBIN* mutant cattle that developed bovine chondrodysplastic dwarfism (BCD) (Figure 1B,C). Studies of mice revealed phenotypes similar to cattle (Figure 1C,E,F), which is the result of reduced, but not eliminated, hedgehog (Hh) signaling. Both EVC and EVC2/LIMBIN proteins have also been shown to form a complex at the bottom of primary cilia. This has led to the classification of EVC as a ciliopathy or disorder of the primary cilia. Because the management of patients with EVC is currently limited to corrective therapy, deepened understanding of disease pathogenesis from animal models raises the possibility of targeted therapy and improved clinical outcomes. 

The review highlights lessons in disease etiology learned from mouse models of EVC with targeted mutations in the causative genes. We will cover: (Section 2) molecular biology of EVC and EVC2/LIMBIN, (Section 3) human disease signs, (Section 4) dysplastic limb development, (Section 5) craniofacial anomalies, (Section 6) tooth anomalies, (Section 7) tracheal cartilage abnormalities, and (Section 8) EVC-like disorders in non-human species.

## 2. Primary Cilium, Ciliopathy, and EVC

Despite longstanding clinical description, it was not until the late 1990s and early 2000s that the two causative genes of EVC (i.e., *EVC* and *EVC2/LIMBIN*) were successfully mapped in human genomes [3,5]. Subsequent molecular studies have demonstrated that both EVC and EVC2/LIMBIN locate and play an important role within the primary cilium [9,10,11,12] (Figure 2A,B). Cilia are rod-shaped microtubule-based surface structures present in nearly all vertebrate cells [13]. Cilia can be categorized into primary cilia and motile cilia, the former of which have nine pairs of microtubule doublets and are immotile, and the latter of which have one more pair of microtubule doublets in the center of the nine pairs of microtubule doublets and can perform rotary movement [13], which is critical for the normal physiology of brain, lung, and sinus [14,15]. Different from motile cilia, which are only present in specific cells, primary cilia are present in nearly all vertebrate cells and they are critical for various aspects of development, post-development homeostasis, and diseases [16]. The unique receptor composition within the primary cilium makes it an important signaling center in vertebrate cells. Congenital mutations in a series of primary ciliary components leads to dysfunction of the primary cilium and results in syndromic disorders (termed ciliopathies) in which organogenesis during embryonic and postnatal development are affected. Studies in the past 15 years have demonstrated that the primary cilia are involved in regulating Hedgehog, G protein-coupled receptors, WNT, receptor tyrosine kinases, and transforming growth factor-beta (TGF-beta)/bone morphogenetic protein (BMP) [16]. Of these, the regulatory function of the primary cilium in Hedgehog signaling induction is best studied. In the absence of the Hedgehog ligand, Hedgehog signaling receptor (i.e., PTCH1) is enriched within the primary cilium, whereas Hedgehog signaling effector protein (i.e., Smoothened, SMO) is expelled out of the primary cilium [17] (Figure 2D). At this time, transcription factor GLI is processed into repressor form for Hedgehog target genes at the ciliary base. The binding of the Hedgehog ligand with PTCH1 leads to a conformation change of PTCH1, which leads to PTCH1 trafficking out of primary cilium and the accumulation of SMO within the primary cilium [17] (Figure 2E). The accumulation of SMO leads to subsequent ciliary trafficking and ciliary tip accumulation in a series of Hedgehog signaling components, including GLI1, GLI2, GLI3, KIF7, and SUFU [18,19,20]. Shuttling in and out of the primary cilium leads to the processing of GLI proteins into activators, which translocate into the nucleus after trafficking out from the primary cilium and function as activators of Hedgehog signaling responsive genes [18] (Figure 2E). EVC and EVC2/LIMBIN form a protein complex with SMO upon induction of Hedgehog signaling and this interaction has been demonstrated to be dependent on the primary cilium [11,12,21]. Based on these, it would be reasonable to expect that the overexpression of wild type *Evc* and *Evc2/Limbin* would lead to increased cellular response to Hedgehog ligand. However, forced expression of *Evc* and *Evc2/Limbin* together in NIH3T3 cells leads to decreased response to Hedgehog ligands [11]. Further studies are needed to delineate the mechanism of how forced expression of *Evc* and *Evc2/Limbin* together leads decreased response to Hedgehog ligands.

Biochemical studies indicate that both EVC and EVC2/LIMBIN are *N*-terminal anchored membrane proteins that form a protein complex and are mutually required for ciliary localization [22]. While EVC and EVC2/LIMBIN share limited homology, the loss of function of one abolishes ciliary localization of the other, which can explain why clinical signs caused by mutations in *EVC* resemble those caused by mutations in *EVC2/LIMBIN* [11,22]. Within primary cilia, the EVC-EVC2/LIMBIN complex is located in a specific zone between the transition zone and Inversin zone [21] as shown in Figure 2D,E. The specific zone in cilia is therefore named the EVC zone. Correct localization into the primary cilium and correct sub-ciliary localization within the primary cilium is critical for the function of EVC and EVC2/LIMBIN. Compared to EVC, the molecular biology of EVC2/LIMBIN is better studied. There are two major consensus sequences located at the *C*-terminus of EVC2/LIMBIN that are required for its precise localization: the FV domain and W sequence (Figure 2C). The FV domain, containing two amino acids, phenylalanine, and valine, is located at the *C* terminal of EVC2/LIMBIN. It is responsible for locating EVC2/LIMBIN to the primary cilium and is conserved in almost all vertebrate, including fish. A disrupted FV domain leads to failed ciliary EVC2/LIMBIN localization and attenuated response to the induction of Hedgehog signaling. The W sequence was initially identified in Weyers acrodental dysostosis, a phenotypically milder form of EVC. A functional FV domain but abolished W domain leads to a dominant negative form of EVC2/LIMBIN, which still locates into primary cilium but has aberrant sub-ciliary localization (i.e., generalized as opposed to restricted to the bottom of cilium) (Figure 2G) [21,23]. Consequently, cellular response to Hedgehog signaling induction is attenuated to a lesser degree in cells with the abolished FV domain. Mechanistic studies indicate that protein EFCAB7 is responsible for tethering the EVC-EVC2/LIMBIN complex to the EVC zone within primary cilium via interaction with EVC2′s W domain [21]. These studies provide insight into the genetics of EVC, particularly regarding mutations identified in *EVC2/LIMBIN* (Figure 2F). In the majority of EVC patients with *EVC2/LIMBIN* mutations, disease is inherited in a recessive manner and failed ciliary EVC2/LIMBIN localization can be attributed to non-functional FV caused by frame shift mutation [6]. Dissimilarly, Weyers acrodental dysostosis has a dominant inheritance pattern, a functional FV domain, a non-functional W domain, and a dominant negative form of the mutated EVC2/LIMBIN protein [4,6] (Figure 2G). In contrast to studies on EVC2/LIMBIN, there are almost no biochemical studies on EVC despite many frame shift mutations identified in human patients [6]. This similarity suggests that, like EVC2/LIMBIN, the *C*-terminus of EVC may likewise be critical for the function of the EVC-EVC2/LIMBIN complex. Experimental evidence is warranted to prove the importance of the *C* terminal in EVC. In addition to the frame shift mutations, there are also missense mutations identified in *EVC* and *EVC2* in human affected by EVC [6]. All mutations identified in *EVC* and *EVC2*/*LIMBIN*, except those in the W domain of *EVC2*/*LIMBIN*, are recessive alleles. There is a recessive allele that is linked to less severe signs in affected individuals [24], prompting an idea that this recessive allele may be hypomorphic. However, it is difficult to make a judgement if any *EVC* or *EVC2/LIMBIN* mutations are hypomorphic alleles through the severity of presented signs because (1) the genetic background in human population is complex; (2) the population size of individuals affected by EVC is very small. Mechanistic studies at molecular levels are needed to determine if mutant alleles are hypomorphic.

It should be noted that multiple lines of evidence suggest that EVC or EVC2/LIMBIN loss of function only partially compromises Hedgehog signaling [9,11,25]. This critical detail is supported by both in vitro molecular studies and in vivo phenotypic studies. For example, *Evc2/Limbin* mutant mouse embryonic fibroblast (MEF) or primary chondrocytes show two thirds of *Gli1* or *Ptch1* expression levels in comparison to control cells [9,11]. This is further supported by comparisons of mRNA levels of *Gli1* and *Ptch1* between multiple tissues between control and *Evc* or *Evc2/Limbin* mutant mice [10,25]. As pointed out by Zhang et al. [25], the partially compromised, instead of largely attenuated, Hedgehog signaling is the likely reason leading to normal lips, palate and neural tube development in humans affected by EVC and in *Evc or Evc2/Limbin* mutant mice [9,10,25], as summarized in Table 1. On the other hand, simply believing that the partially compromised Hedgehog signaling is the reason leading to the abnormal phenotypes in mutant mice or signs in humans may lead to an inaccurate understanding of the pathological mechanism. On the other hand, it is still not known if as ciliary components, EVC and EVC2/LIMBIN are involved in the ciliary regulation of other signaling pathways, and how affected non-Hedgehog signaling contributes to the pathological mechanism of EVC is remains elusive.

## 3. Overview of Human Signs

Classic signs associated with EVC include polydactyly, congenital morbus cordis or heart disease, chondrodysplasia, and ectodermal dysplasia [1]. However, this tetrad is not present in all cases and there are reported degrees of severity [4]. Patients require an interdisciplinary treatment strategy due to symptom variability and prognosis which is ultimately dependent upon the extent of cardiac anomalies [26,27,28]. As a result, reports of adult patients with EVC are less common, though this may represent a bias towards pediatric diagnosis, based on apparent disease signs (e.g., polydactyly and chondrodysplasia) [28,29]. Individuals surviving beyond the tenuous neonatal period may live a normal lifespan and tend to have normal emotional and intellectual development, though there are reported instances of impairment [30].

Polydactyly, or the presence of extra digits, is the most conspicuous sign of EVC at birth [1,4]. Hands may have between 6–7 digits with extra fingers always present on the ulnar side (i.e., postaxially). Toes, though typically normal in number, may be irregularly aligned; both sets of appendages have dysplastic nails due to the associated ectodermal dysplasia. Patients exhibit progressive distal shortening of the fingers with underdevelopment of distal phalanges that may include non-functional distal metacarpals. Additional digits can be surgically removed; however, this does not improve the “plump” appearance due to the underlying shortened bones.

Congenital heart disease is present in 50–60% of EVC patients and is the most significant cause of morbidity [4,26,27,28,31]. Septal defects range in severity with the most severe being cor biloculare or a two chambered heart. Venous anomalies concurrent with either atrial septal defects (ASD) or ventricular septal defects (VSD) may contribute to death via pulmonary hypotension. Accurate diagnosis and management of cardiac abnormalities are therefore necessary for improved disease prognosis.

Chondrodysplasias are disorders of bone growth that manifest as shortened stature. Though potentially non-apparent at birth, disproportionate dwarfism in patients with EVC is the result of long bones that are more shortened at their distal ends [1,32,33]. Patients have a normal length torso but dysplastic primary ossification centers in the paired bones of the arms and legs as well as the digits. Shortened ribs also result in a small chest with a keel-like breastbone that may not allow for proper lung expansion [31]. Musculoskeletal problems, such as genu valgum (knock knee) or talipes equinovarus (club foot), may restrict mobility and require orthopedic intervention (e.g., surgery or casting) to correct [4,33]. Interphalangeal ankylosis, most often seen in the distal metacarpals, may limit dexterity necessitating lifestyle adaptation and/or corrective surgery.

Ectodermal dysplasia affects ectodermal derived tissues, such as the hair, skin, nails, and teeth. Compared to other types of ectodermal dysplasia, hair and skin abnormalities are mild in EVC patients and occur in only about 1/3 of cases [4,34]. As previously mentioned, nails of both the hands and feet of patients with EVC are malformed with those of the hands, and especially supernumerary digits, being more affected. However, all patients with EVC have a dental phenotype that includes hypoplastic enamel, congenitally missing teeth (i.e., hypodontia), abnormally shaped teeth (e.g., taurodontism or bulbous teeth), and premature eruption and exfoliation [28,35,36]. Abnormal craniofacial morphology may not be apparent at birth but includes a trend towards mandibular prognathism, maxillary deficiency, skeletal open bite, and prominent frontal bossing that gives patients a concave-shaped lateral profile. While not at an inherently increased risk of caries, the loss of oral function often requires extensive prosthodontic rehabilitation (e.g., dentures and implants) starting from a young age [37,38]. Depending on the severity of the craniofacial phenotype, patients with EVC may also benefit from treatment by speech–language pathologists.

In addition to the classic signs of EVC, less common findings include genital abnormalities and strabismus [2,33]. Abnormal urethral location (i.e., hypospadias) and undescended testicles (i.e., cryptorchidism) in males is rare, though the former is common in other forms of ectodermal dysplasia. Strabismus, a vision condition characterized by the misalignment of the of the eyes, is due to problems associated with the muscles that move the eyes (i.e., extraocular muscles). These muscles develop with influence from cranial neural crest cells, thereby suggesting an expanded role of EVC2/LIMBIN in the craniofacial region (see Section 5: Craniofacial Phenotype) [39]. Both genital and eye muscle abnormalities are non-life threatening and may be corrected with surgery.

## 4. Limb Phenotypes

Dwarfism or shortened stature is one of the most typical symptoms present in patients with EVC. Adult patients with EVC usually fall into the height range of 110 cm to 155 cm (−2 to −4.5 standard deviation scores) [40]. However, growth hormone deficiency is not consistently observed in these patients and growth hormone treatment in pediatric patients does not achieve consistent efficacy [40]. This suggests that dwarfism in patients with EVC is not likely due to deficiency or abnormal growth hormone levels. Instead, dwarfism in EVC is mainly characterized by the disproportional distal shortening of arms and legs, whereas the trunk size of patients remains unchanged [1]. The pathological mechanism leading to the dwarfism in patients with EVC therefore lies in abnormal appendicular bone development.

Appendicular bone elongation during development occurs through endochondral ossification, a process in which chondrocyte proliferation and maturation in cartilage play vital roles in determining the final length of bones in the arms and legs [41,42]. Appendicular bone development is initiated from the condensation of mesenchymal cells and subsequent differentiation to chondrocytes [41]. Chondrocytes then undergo a series of proliferation and maturation steps, which allows for the formation of the primordial cartilage of appendicular bone. Chondrocyte proliferation and maturation are well orchestrated by a series of locally produced factors that ensure the correct length and shape of each skeletal element. Previous studies have demonstrated that many signaling pathways play important roles in regulating chondrocyte proliferation and maturation (Figure 3A). For example, Fibroblast Growth Factor (FGF) signaling, mediated by FGF18 produced in the perichondrium, is involved in this regulatory network [43,44,45]. In the growth plate, FGF signaling inhibits chondrocyte proliferation through STAT1-mediated p21 expression and inhibits chondrocyte differentiation through MEK/PERK-mediated signaling [44,45]. Additionally, Indian Hedgehog and parathyroid hormone-related protein (PTHrP) signaling also play important roles in the regulation of chondrocyte proliferation, differentiation, and maturation. Indian Hedgehog is expressed by the pre-hypertrophic chondrocytes and stimulates PTHrP expression in chondrocytes at the distal end of the growth plate. PTHrP promotes chondrocyte proliferation and prevents them from directly progressing to pre-hypertrophic differentiation [46]. Once proliferating chondrocytes are far away enough from the source of PTHrP (i.e., chondrocytes at the distal end of the growth plate), they differentiate into pre-hypertrophic chondrocytes, start expressing Indian Hedgehog, and then further mature into hypertrophic chondrocyte cells [41,47] (Figure 3A).

Compromised cellular response to Hedgehog signaling seen in *Evc* or *Evc2/Limbin* loss of function mouse models has been originally speculated as the pathological mechanism leading to dwarfism in EVC [10,11,21]. Based on the current understanding, decreased Hedgehog/PTHrP signaling in the chondrocyte should lead to subsequently decreased proliferation and abnormal maturation of chondrocytes in cartilage of the appendicular skeleton. Indeed, studies using mouse models confirmed the decreased Hedgehog signaling and PTHrP expression in the growth plate of both *Evc* and *Evc2/Limbin* mutant mice. However, Zhang et al. [25] presented surprising evidence suggesting that compromised Hedgehog signaling is not the reason leading to dwarfism; chondrocyte specific deletion of *Evc2/Limbin* leads to very mild dwarfism at embryonic day E18.5 and leads to no dwarfism at postnatal day P21 [25].

Subsequent studies of abnormal signaling in the *Evc2/Limbin* mutant growth plate by Zhang et al. [25] implicate an elevated FGF18 secretion from the perichondrium as a critical contributor to the dwarf phenotype (Figure 3B) [25]. Chondrocyte-specific (note: non-perichondrium) deletion of *Evc2/Limbin* results in a mild dwarf phenotype despite compromised Hedgehog signaling along with unaffected FGF signaling at both embryonic and postnatal stages. On the other hand, concurrent *Evc2/Limbin* deletion in both chondrocytes and perichondrium results in a severe dwarfism with compromised Hedgehog signaling and elevated FGF signaling at embryonic and postnatal stages [25] suggesting that increased FGF18 production in the mutant perichondrium is critical to inhibit chondrocyte proliferation and leads to dwarfism (Figure 3B, red arrow). Though it is still unresolved how *Evc2/Limbin* loss of function within the perichondrium leads to elevated FGF18 expression, the studies from Zhang et al. [25] highly suggest that elevated FGF signaling is a potential druggable target for dwarfism in patients with EVC [25]. Elevated FGF signaling in the growth plate is the pathological mechanism leading to another form of dwarfism (i.e., achondroplasia) and a soluble form of FGFR3 (sFGFR3) was recently evaluated in a clinical trial for treating this dwarfism [48]. The similar pathological mechanism for dwarfism in achondroplasia and EVC suggests that sFGFR3 may be a viable therapeutic for treating dwarfism in patients with EVC.

## 5. Craniofacial Phenotype

The concave facial profile seen in patients with EVC suggests morphologic differences in the midfacial and skull base regions. Mice highly express both *Evc* and *Evc2/Limbin* within these areas and *Evc* and *Evc2/Limbin* deficient mice show craniofacial phenotypes that recapitulate human signs [9,10,11,49,50,51]. *Evc* is expressed in the maxillary and mandibular processes from embryonic day E11.5 and detectable in the nasal septum at E15.5 [10]. *Evc2/Limbin* at E15.5 is similarly expressed within the mandible and maxilla as well as in the nasal, premaxilla, cranial sutures, and spheno-occipital synchondrosis [8,50]. Though no obvious craniofacial differences are discernible at embryonic time points, the structures listed above collectively represent cartilage-derived elements of the braincase and facial skeleton (i.e., neurocranium and viscerocranium), thereby providing insight into postnatal disease pathogenesis.

Many elements within the craniofacial region, specifically the neurocranium and viscerocranium, arise from neural crest cells (NCCs), a special population of cells that maintain their identity despite divergence from the ectoderm in early gastrulation [52]. Abnormal craniofacial morphology (i.e., midfacial depression) in both global and NCC-specific *Evc2/Limbin* deficient mice (referred to as *Evc2/Limbin*-KO and *Evc2/Limbin*-cKO, respectively) is a consequence of abnormalities within NCC-derived elements of the neurocranium and viscerocranium [50,51,53,54]. As with patients with EVC, midfacial depression or retrusion in mice becomes more pronounced with age and can be attributed to positional differences between the nasal bone, jaws, and cranial base [51,53]. Though this suggests a critical, NCC-specific role for *Evc2/Limbin* in determining postnatal craniofacial morphology, it does not indicate which structure (i.e., maxilla vs. skull base) is responsible for the resultant phenotype.

Cephalometric analysis is required to determine the etiology (i.e., maxilla vs. skull base deficiency) of midfacial retrusion in patients with EVC. Although such analyses are rare within the literature, findings from different *Evc2/Limbin*-cKO mouse models implicate the role of *Evc2/Limbin* in the anterior skull base (Figure 4A,C). Though both Cre “drivers” (i.e., Wnt1-Cre and P0-Cre) used to create *Evc2/Limbin*-cKO mice are expressed throughout the jaws and facial skeleton, Wnt1-Cre-based recombination efficiency in the anterior skull base is much higher (100% vs. 50% for Wnt1-Cre and P0-Cre, respectively), results in worse midfacial deficiency, and causes greater skull base defects that in their *Evc2/Limbin*-P0-cKO counterparts [53]. This is significant as the skull base is a midline structure that connects the posterior of the head with the facial region and aberrant skull base morphology has been found to contribute to other syndromic facies.

A part of the neurocranium, the skull base is formed by endochondral ossification and consists of three segments connected by two growth plates or synchondroses, all mirrored and fused across the midline. Though skull base expression of *Evc* and *Evc2/Limbin* is lower than in the facial skeleton, all animal models of EVC (i.e., *Evc*-KO, *Evc2/Limbin*-KO, and *Evc2/Limbin*-cKO) have discernible differences in skull base size, shape, and bone quality [11,49,50,53]. Despite differences in facial gross morphologies, sizes and shapes of calvarial bones (nasal, frontal and parietal) are comparable between mutants and controls (Figure 4B). Morphometric differences in *Evc2/Limbin*-cKO mice were most apparent in the anterior vs. posterior skull base (i.e., presphenoid and basisphenoid vs. basioccipital) where they, as with *Evc*-KO mice, also showed the premature fusion of the intrasphenoidal synchondrosis (ISS) between these two parts [49,53] (Figure 4C). As expected, no differences were seen in the non-NCC-derived basioccipital at the posterior skull base (Figure 4C) and spheno-occipital synchondrosis (SOS) remains open in the *Evc2/Limbin*-cKO. The collective evidence suggests that *Evc2/Limbin* genes affect postnatal craniofacial morphology by maintaining the ISS and allowing for anterior skull base elongation. This critically explains why patients with EVC present progressively worsening midfacial depression (Figure 4).

## 6. Tooth Phenotype

Embryonic tooth development requires a series of morphological differentiations and the interaction between the dental epithelium and mesenchyme. Ameloblasts and odontoblasts arise from the dental epithelium and mesenchyme, respectively, and secrete matrix proteins necessary for the formation of enamel and dentin. Despite this crosstalk, ameloblast and odontoblasts have different developmental origins and, respectively, arise from ectodermal or neural crest cells [55]. The recapitulation of human dental abnormalities in animal models has revealed expanded and cell-specific roles of *Evc* or *Evc2/Limbin* in disease pathogenesis.

Global disruption of *Evc* or *Evc2/Limbin* and NCC-specific disruption of *Evc2/Limbin* in mice all result in a range of dental phenotypes of the likes seen in patients with EVC. This includes enamel hypoplasia, hypodontia (i.e., congenitally missing teeth), and progressively abnormal tooth morphology (e.g., small, conically shaped, fused, and/or short rooted) [9,10,51,56]. Phenotypic similarity between global and NCC-specific *Evc2/Limbin* deficient mice importantly suggests that *Evc2/Limbin* expression in the NCC-derived dental mesenchyme determines tooth morphology and enamel development. This means that tooth and enamel phenotypes seen in patients with EVC are secondary results, driven by misexpression in NCC-derived cells.

## 7. Tracheal Cartilage Phenotypes

Unlike other symptoms of EVC that become more apparent throughout childhood, airway obstruction or collapse is an immediate and life-threatening condition that may have been overlooked and underreported for a long time. Neonatal death associated with collapsed airway has been documented in infant patients with EVC. Biopsies indicate hypomorphic tracheal cartilage that is unable to support the tracheal epithelia during the initiation of breathing [1]. The insufficient cartilaginous support of the airway at laryngeal, tracheal, or bronchi level condition is termed laryngotracheobronchomalacia and may also lead to stridor or noisy breathing in infant patients with EVC [1]. Similarly, neonatal death was reported in *Evc2/Limbin* mutant mice [9]. Whether the neonatal death is due to defective cartilage structure will be worth investigating.

## 8. EVC-Like Disorders in Other Non-Human Species

Examples of dwarfism or small stature resulting from a medial or genetic condition are not uncommon throughout the animal kingdom. However, while news headlines are dominated by the discovery of insular dwarf species and deliberate selection has led to the development of numerous toy and true “dwarf” animal varieties, chondrodysplastic cases attributable to mutations in animal orthologs (specifically *EVC2/LIMBIN*) are restricted to two breeds of cattle. However, recent comparative studies have the potential to spark interest in the etiology of EVC-like disorders and reveal the evolutionary significance of EVC2/LIMBIN in other species.

The Japanese brown is a horned cattle breed esteemed for its use in beef production, known as wagyu. Starting in the late 1980s, calves from a specific region of Japan were born with a type of dwarfism (i.e., ateliosis) characterized by disproportionately short limbs and joint abnormalities [57]. Termed bovine chondrodysplastic dwarfism (BCD), this unusual phenotype was inherited in an autosomal recessive fashion and was later found to be caused by either deletion or frameshift mutations in a gene (i.e., *LIMBIN*) located on chromosome 6 [57,58]. *LIMBIN* was subsequently identified as an ortholog of human *EVC2* despite the lessened severity of symptoms in bovine vs. human patients [8]. Identification and targeted manipulation of the murine ortholog laid the groundwork for the generation and phenotypic characterization of non-human models of EVC (see previous sections) that have been invaluable in our mechanistic understanding of disease development.

A similar form of disproportionate dwarfism in Tyrol grey cattle was reported in the early 21st century. The Tyrol grey is a horned, dual purpose breed of Austrian and Italian origin considered endangered due to its small population (<4000 individuals vs. >20,000 for Japanese brown) [59,60]. Affected calves could be traced back to a single ancestor and displayed the characteristic small stature and short limbs seen in both Japanese BCD and human EVC examples [59]. In addition to growth plate abnormalities in long bones indicative of dysplastic chondrocyte function, some Tyrol BCD specimens also displayed acetabular joint laxity, urogenital abnormalities, and cardiac defects reminiscent of human patients with EVC [59,61]. However, orofacial, ectodermal, and digit findings in these cattle were unremarkable and appendages were shortened in a proximal vs. distal manner as indicated by the shortened femur and humerus. Though genetic analyses identified an *EVC2/LIMBIN* mutation (i.e., deletion) at a different location from their Japanese counterparts, these cattle provided an animal model of an EVC-like disorder and expanded insight into the role of EVC2/LIMBIN in skeletogenesis.

Penguins are flightless birds that have undergone numerous morphological adaptations suited to an aquatic lifestyle. These include modification of the upper appendages into robust, flipper-like wings and pronounced shortening of femoral and lower extremity length, characteristics that are superficially like those seen in patients with EVC (see previous sections). Genetic analysis of two penguin species (i.e., *Pygoscelis adeliae* and *P. forsteri*) surprisingly revealed five penguin-specific amino acid changes in EVC2/LIMBIN, the highest amongst all analyzed limb-related genes [62]. An additional amino acid change was seen in the penguin EVC ortholog. These results suggest important evolutionary roles of *EVC* and *EVC2/LIMBIN* orthologs due to their conserved effects on vertebrate (e.g., mammalian and avian) limb development.

## 9. Conclusions

It has been 80 years since the first report of EVC, 13 years since the report of *Evc* mutant mice, and 5 or 8 years since two independent reports of two *Evc2/Limbin* mutant mouse models. Within this short time, genetic mouse models have allowed great insight regarding the pathogenesis of EVC, expanded both breadth and depth of knowledge, and provided potential insight into therapeutic solutions for different aspects of EVC. Beyond patient care, *Evc2/Limbin* mutant mice have become a precious genetic tool for propelling our current understanding in molecular, cellular, and development biology.

## Figures and Tables

**Figure 1 jdb-08-00025-f001:**
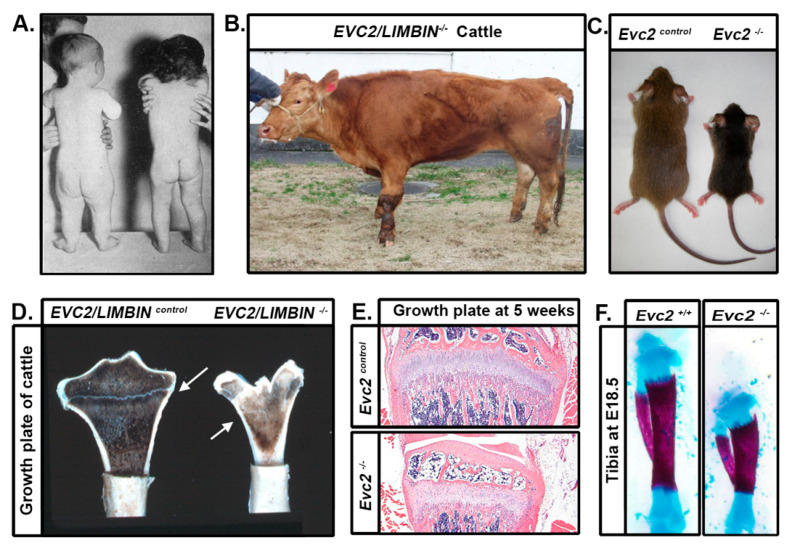
Individuals affected by EVC, *LIMBIN* mutant cattle, and *Evc2/Limbin* mutant mice share similar dwarfism. (**A**) A 20-month-old infant affected by EVC syndrome (left) and a 20-month-old infant without EVC syndrome (right) © BMJ. The infant with EVC syndrome has un-proportional shortened legs. (**B**) A *LIMBIN* mutant cattle bears apparent short legs. (**C**) *Evc2/Limbin* mutant mice with littermate controls exhibit a smaller body size at 5 weeks old. (**D**) *LIMBIN* mutant cattle have decreased size of growth plate (marked in white arrow) in appendicular bones. (**E**) The growth plate of tibia from *Evc2/Limbin* mutant mice demonstrate shorter and disorganized structure at 5 weeks old compared with control mice. Bars indicate 200 um. (**F**) Tibiae from E18.5 mouse embryos were stained with alcian blue for cartilage and alizarin red for bone. Tibiae from *Evc2/Limbin* mutant embryos are shorter than those in control littermates, although body size in these two groups are similar at this stage.

**Figure 2 jdb-08-00025-f002:**
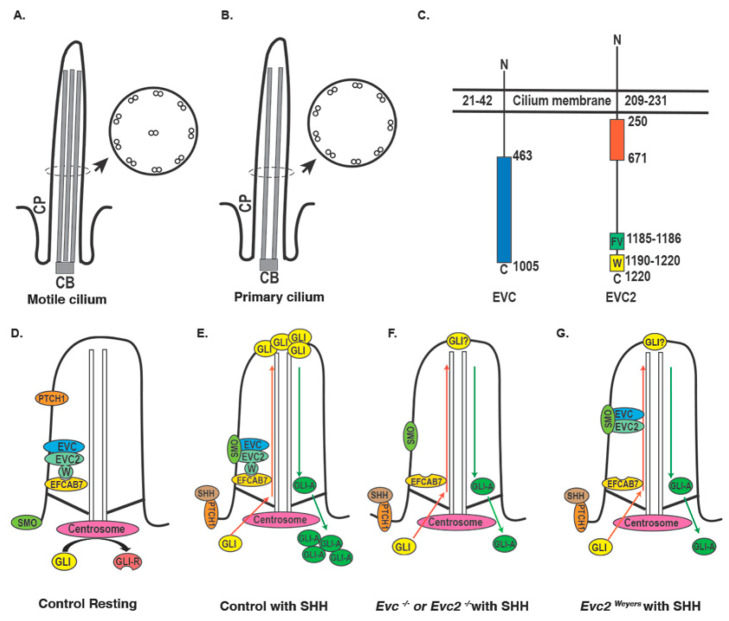
Primary cilium, EVC and EVC2/LIMBIN proteins, and the mechanism of EVC-EVC2/LIMBIN in regulating Hedgehog signaling within the primary cilium. Diagrams of the motile cilium (**A**) and the primary cilium (**B**) are shown. CB, cilium base; CP, cilium pocket. (**C**). Structures of the EVC and the EVC2/LIMBIN are shown. The EVC and EVC2/LIMBIN are *N* terminal anchored proteins. Blue box in the EVC and orange box in the EVC2/LIMBIN indicate domains for interaction with each other. Green box indicates the domain for the ciliary localization of the EVC2/LIMBIN and the yellow box indicates the domain for localization of the EVC2/LIMBIN at the EVC zone. Numbers indicate the numbers of amino acids in each protein. (**D**) EVC-EVC2/LIMBIN complexes are localized at the bottom of cilia by tethering to EFCAB7 through the W domain in EVC2/LIMBIN. In the absence of Hedgehog ligand, PTCH1 resides within the primary cilium, and GLI proteins are processed to the repressor form (GLI-R) at the centrosome. (**E**) In the presence of Hedgehog ligand, binding of the ligands with PTCH1 leads to exclusion of PTCH1 out of the primary cilium, which allows SMO to enter the primary cilium. Within the primary cilium, SMO interacts with EVC-EVC2/LIMBIN at the bottom of the primary cilium, which allows GLI trafficking into the primary cilium and accumulation at the tip of the primary cilium. After entering the primary cilium, GLI are processed to the activator form (GLI-A). GLI activators exit the primary cilium and translocate into the nucleus to activate Hedgehog responsive genes. (**F**) EVC or EVC2/LIMBIN loss of function leads to absence of EVC-EVC2/LIMBIN complexes within the primary cilium. When Hedgehog signaling is activated, SMO still moves into the primary cilium, but without EVC-EVC2/LIMBIN complexes, SMO cannot lead to full activation of GLI. (**G**) In Weyers form of mutant cells, EVC-EVC2/LIMBIN complexes cannot be restricted at the bottom of primary cilium due to no interactions with EFCAB7 caused by loss of the W domain in EVC2/LIMBIN, thus EVC-EVC2/LIMBIN-SMO complex cannot lead to full activation of GLI.

**Figure 3 jdb-08-00025-f003:**
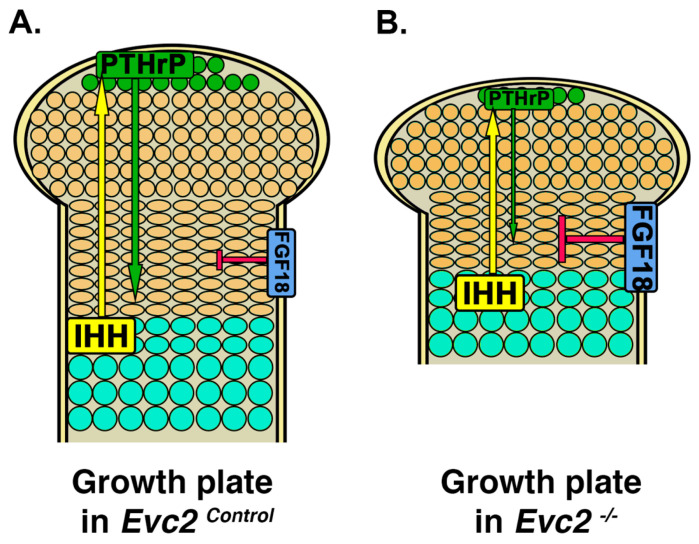
Elevated FGF signaling is critical for the pathogenesis of the dwarfism developed in EVC. (**A**) In control growth plate, both Hedgehog signaling (yellow-green feedback loop) and FGF signaling (red) work on chondrocytes to ensure regulated proliferation and maturation. (**B**) In *Evc2/Limbin* mutant growth plate, moderately decreased Hedgehog signaling due to *Evc2/Limbin* loss of function within only chondrocytes moderately contributes to the pathogenesis of dwarfism, whereas elevated FGF signaling due to loss of *Evc2/Limbin* within perichondrium critically contributes the pathogenesis of the dwarfism. Green, resting chondrocytes; gold sand, proliferating chondrocytes; aqua, hypertrophic chondrocytes, IHH, Indian Hedgehog ligand.

**Figure 4 jdb-08-00025-f004:**
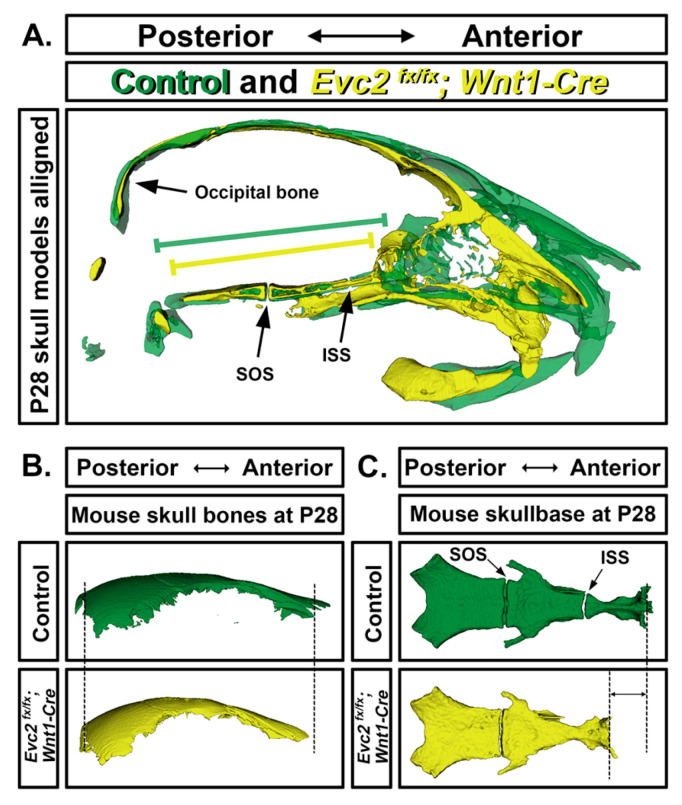
A shortened skull base leads to mid-facial defects in *Evc2/Limbin* mutant mice. (**A**) Surface models of the mid-line regions were generated based on the micro-CT scans of controls (green) and *Evc2/Limbin* mutants (yellow). Two models were then superimposed at the occipital bones of the skull. Green and yellow lines are spanning the entire regions of the skull bases in control and mutant, respectively; black arrows indicate the intersphenoidal synchondrosis (ISS) and the spheno-occipital synchondrosis (SOS) in skull base. (**B**) Models were generated from the skull region containing nasal, frontal and parietal bones from micro-CT scans of control and mutant. No apparent defects were observed in *Evc2/Limbin* mutants in comparing to controls. (**C**) Models were generated from the skull base from micro-CT scans of controls and mutants. Apparent shortened anterior parts of skull bases from *Evc2/Limbin* mutants are observed in comparison to controls, whereas the posterior part of the mutant skull bases remains the same length with the controls. Black arrows indicate the ISS and the SOS.

**Table 1 jdb-08-00025-t001:** Clinical signs of EVC and phenotypes of *Evc*, *Evc2/Limbin* mutant mice, and mice with Hedgehog signaling defects.

Anatomical Locations	EVC	Weyers Acrodental Dysostosis	*Evc* Mutant Mice	*Evc2/Limbin* Mutant Mice	Hedgehog Signaling Defects in Mice
Limb	Short	Short (mild)	Short	Short	Short
Craniofacial	Normal	Normal	Normal	Normal	Cleft lips, cleft Palate
Neural tube	Normal	Normal	Normal	Normal	Open neural tube
Digit	Postaxial polydactyly	Postaxial polydactyly	Postaxial polydactyly	Postaxial polydactyly	Preaxial polydactyly (?)

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
