# Peer review of "Molecular and Cellular Pathogenesis of Ellis-van Creveld Syndrome: Lessons from Targeted and Natural Mutations in Animal Models"

_jdb, 2020, doi:10.3390/jdb8040025_

Round 1

Reviewer 1 Report

The manuscript ID: jdb-926196 – Review titled: “Molecular and Cellular Pathogenesis of Ellis-van Creveld Syndrome: Lessons from Targeted and Natural Mutations in Animal Models” starts from the Ellis-van Creveld syndrome (EvC) which is a rare human congenital disease characterized by remarkable skeletal dysplasia such as short limbs, ribs and polydactyly, and orofacial anomalies.

Then it extends the analysis to animal models for EvC providing and highlighting the utility of such animal models to deepen our understanding into disease pathogenesis.

This review highlights the utility of animal-based studies of EvC by summarizing: (1) molecular biology of EVC and EVC2/LIMBIN, (2) human disease signs, (3) dysplastic limb development, (4) craniofacial anomalies, (5) tooth anomalies, (6) tracheal cartilage abnormalities, and (7) EvC like disorders in non-human species.

The manuscript is well written and well detailed giving an overview of the current knowledge about the EvC and EvC like disorders in human and non-human species and molecular biology of EVC and EVC2/LIMBIN.

The manuscript is ready for publication after minor revisions.

Minor revisions

Add MIM number for the all the human syndromes reported in the manuscript.

Please specify since the beginning if they are dominant or recessive disorders.

Use the same nomenclature of OMIM throughout the text i.e EVC instead of EvC for both human and mouse.

Please use the correct nomenclature for Weyers acrodental dysostosis (WAD and not WCH).

All gene names should be in italic.

In all figures, you should avoid (Louise et al.).

In the Introduction, lines 42-50 please add some references for animal models of EVC and EVC/EVC2 proteins function.

In figure 1, it would be important to add also a human phenotype for the dwarfism.

The review highlights several lessons, briefly summarized from 1 to 7, however these numbers do not coincide with those reported and expanded later. Please check and adjust accordingly.

Figure 2 is too small and the name of some proteins, i.e. EFCAB7 are not readable. Please adjust it.

In the description of the EVC and EVC2 proteins, it would be helpful to have a schematic figure depicting linear arrangement of both proteins’ domains, which are those affected in the related syndromes and what they bind to.

In Table 1, it would be better to move the column “Hedgehog signaling defects in mice” as the last one. It would be important to add also a column for patients with WAD.

Add a reference for sentence at page 6 lines 243-245.

At page 6 lines 249, 254, 262, please add a date for Zhang et al. since more than one paper is present in the reference list for it.

In figure 3 legend, please specify the acronym for IHH. Furthermore, the names of the signaling pathways are not easily readable.

In figure 4, fx/fx is not well readable in several points of the figure. Please adjust.

In the last point, 8, “EVC-like disorders in non-human species”, the authors discuss a paragraph about the mouse models of EVC, but these models are widely discussed and detailed in previous sections, so probably they should consider moving this point after the human disease signs point 2, before points number 3-6.

For References, please use the Reference list and citations style guide for MDPI journals.

Author Response

- Add MIM number for the all the human syndromes reported in the manuscript.

OMIM entries added in lines 10, 23, and 35.

- Please specify since the beginning if they are dominant or recessive disorders.

Updated in lines 23 and 34.

- Use the same nomenclature of OMIM throughout the text i.e EVC instead of EvC for both human and mouse.

Corrected throughout.

- Please use the correct nomenclature for Weyers acrodental dysostosis (WAD and not WCH).

Corrected to WAD in line 34 and 35

- All gene names should be in italic.

All have been changed as the reviewer’s request.

- In all figures, you should avoid (Louie et al.).

We have changed according to the reviewer’s suggestion.

- In the Introduction, lines 42-50 please add some references for animal models of EVC and EVC/EVC2 proteins function.

We have added more reference as the reviewer requested, now in lines 45-46.

- In figure 1, it would be important to add also a human phenotype for the dwarfism.

We have made changes according to the reviewer’s suggestion in Fig 1. A permission letter from BMJ Publishing Group for the use of the image is obtained and attached.

- The review highlights several lessons, briefly summarized from 1 to 7, however these numbers do not coincide with those reported and expanded later. Please check and adjust accordingly.

Updated lines 54-58 to correspond to section numbers and content (i.e. section 3 and human signs)

- Figure 2 is too small and the name of some proteins, i.e. EFCAB7 are not readable. Please adjust it.

The font size of EFCAB7 was enlarged as suggested.

- In the description of the EVC and EVC2 proteins, it would be helpful to have a schematic figure depicting linear arrangement of both proteins’ domains, which are those affected in the related syndromes and what they bind to.

We have added a diagram to show case the structures of EVC and EVC2 in Fig 2. However, until today, there is very limited knowledge available on the domains in EVC and EVC2 proteins, which cannot explain any missense mutations identified. Currently, the critical domains at EVC2 C terminal can explain the loss of function in many frame shift mutations. This is discussed in the text (lines 148-154).

- In Table 1, it would be better to move the column “Hedgehog signaling defects in mice” as the last one. It would be important to add also a column for patients with WAD.

Table is updated as suggested.

- Add a reference for sentence at page 6 lines 243-245.

References are added as suggested, now in lines 275-278.

- At page 6 lines 249, 254, 262, please add a date for Zhang et al. since more than one paper is present in the reference list for it.

Immediate citation of reference is added in these places, now in lines 289, 293, 296.

- In figure 3 legend, please specify the acronym for IHH. Furthermore, the names of the signaling pathways are not easily readable.

IHH is now spelled out in the figure ligand for Fig 3. The names of the signaling pathways are enlarged according to the reviewer’s suggestion.

- In figure 4, fx/fx is not well readable in several points of the figure. Please adjust.

They have been enlarged as the reviewer suggested.

- In the last point, 8, “EVC-like disorders in non-human species”, the authors discuss a paragraph about the mouse models of EVC, but these models are widely discussed and detailed in previous sections, so probably they should consider moving this point after the human disease signs point 2, before points number 3-6.

We respectfully disagree with moving the section since it is meant to focus on non-mouse models of EVC beyond the murine insights discussed in sections 2 and 4-7. We have renamed the section “EVC-like disorders in other non-human species”

- For References, please use the Reference list and citations style guide for MDPI journals.

We have made changes according to MDPI style.

Reviewer 2 Report

Thanks for the interesting manuscript. EvC is so rare, that I have not met the patients, but it was interesting to read about all the possible connections and biological background.

Small suggestions/questions: 

  • A bit of more general basic biology about the cilia would be good if you can fit it in. EM/ cell architecture pictures or explaining the function of cilia in larger perspective, perhaps. 
  • Fig 2A SMO (outside the picture) would be good to see.
  • What do we know about SHH more drastic phenotypes, e.g.holoprosencephaly in relation to EvC/EvC2? Does SHH signaling go through non-cilia route as well?
  • Does EvC/EvC2 mutation type have an effect on the phenotype? What does overexpression do in cells or animals? Are the hypomorphic alleles or somatic mosaic patients? 

Author Response

Reviewer 2 Comments

- A bit of more general basic biology about the cilia would be good if you can fit it in. EM/ cell architecture pictures or explaining the function of cilia in larger perspective, perhaps.

We have updated the text (lines 74-78) and figure 2 according to the reviewer’s suggestion.

- Fig 2A SMO (outside the picture) would be good to see.

We have added SMO to Fig 2 according to the reviewer’s suggestion.

- What do we know about SHH more drastic phenotypes, e.g.holoprosencephaly in relation to EvC/EvC2? Does SHH signaling go through non-cilia route as well?

The more drastic SHH phenotypes in relation to EVC/EVC2 is well discussed by previous studies (10.1371/journal.pgen.1006510). We have added more discussion in this aspect (lines 176-178). As the reviewer pointed, Hedgehog signaling does go through non-cilia route as well. However, we did not discuss this issue because currently there is no information if Evc/Evc2 loss of function potentially affect Hedgehog signaling through non-cilia route.

- Does EvC/EvC2 mutation type have an effect on the phenotype? What does overexpression do in cells or animals? Are the hypomorphic alleles or somatic mosaic patients? 

We have added more discussions on the EVC/EVC2 mutation type on the human signs (lines 165-166). Currently there is only one studies carried out on overexpression of EVC and EVC2, which is discussed in lines 103-107. A report describing a possible hypomorphic allele is mentioned in lines 166-168. All reports for genetic alterations with pedigree exclude a possibility of somatic mosaic mutations so far, however, there is not enough information to fully rule out somatic, mosaic patients. This fact is discussed in line 166-172.

Reviewer 3 Comments

- Figures. "Louie et al." at the base of figures. The authors may
want to explain if this is a reference or if they want to indicate that
this is an original figure made for this article.

We have removed them.

- Figure2. I suggest to place SMO in the membrane as it is a protein that belongs to the seven-transmembrane G protein-coupled receptor superfamily.

We have made revised figure as the reviewer suggested.

- In addition to Tompson et al., there are two other manuscripts on EVC and EVC2 mutations including Weyers cases which the authors might consider to include in their review: Valencia et al. Hum Mut 2009 and D´Asdia et al  Eur J Med Genet 2013. Valencia et al.  also showed for the first time that ectopic expression of Weyers mutations impairs hedgehog signaling."

We have cited these references as the reviewer suggested.